# Experimental Parametric Study of a Functional-Magnetic Material Designed for the Monitoring of Corrosion in Reinforced Concrete Structures

David Souriou [1,*], Sima Kadkhodazadeh [2], Xavier Dérobert [1], David Guilbert [2] and Amine Ihamouten [1]

[1] Laboratory for Modelling, Experimentation and Survey of Transport Infrastructures (LAMES), Department of Materials and Structures (MAST), University Gustave Eiffel, F-44344 Bouguenais, France

[2] Department Laboratory of Angers (DLan), Cerema Ouest, F-49136 Les Ponts de Cé, France

\* Correspondence: david.souriou@univ-eiffel.fr

**Abstract:** The presence of aggressive agents (such as chloride ions brought by seawater) in reinforced concrete structures is responsible for the corrosion of the steel rebars. A Structural Health Monitoring technology is developed as a new passive preventive method that would allow for the detection of and for the ability to follow the presence of chloride ions in the cover concrete of reinforced concrete. This technology, referenced as Functional Magnetic Material (FMM), consists on the measurement with an external interrogator of a Magnetic Observable (MO), partially shielded by a patch and corrodible by chloride ions. This paper presents the results of a parametric experimental study, allowing the validation of the concept of this technology, by highlighting the variation of the MO while considering the geometry and the corrosion level of the patch (based on its Relative Mass Loss—RML), as well as the distance between the samples and the interrogator. The results show that the MO of the FMM significantly varies with the increase in the RML of the patch. A 10%-RML for the patch is sufficient for detecting a variation of the MO of the FMM, and the relative variations of the MO are strongly dependent on the distance between the FMM and the magnetometer, as well as the patch's thickness.

**Keywords:** corrosion; magnetic flux density; non-destructive technique; ferromagnetic materials; chloride ions; Structural Health Monitoring; Civil Engineering



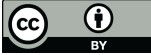

## 1. Introduction

In the domain of Civil Engineering, the corrosion of steel rebars, induced by chloride ions, is one of the most current issues of reinforced concrete (RC) structures, mainly in marine environment [1–3]. The diffusion of chloride ions, brought by seawater into the porous network of the cover concrete area, causes the formation of corrosion products on steel rebars, where volumetric expansion will initiate cracks in the structure [4,5], which finally increases the risks of collapsing.

Nowadays, for safety, maintenance, and economical purposes, Non Destructive (ND) methods are widely used by infrastructure managers to detect the defects and evaluate the pathology caused by the corrosion of rebars in order to estimate the structure's service life through durability indicators [6]. For example, UltraSonic Waves (USW) are used to evaluate the physical characteristics of concrete, by the propagation of a compressive mechanical wave within the concrete, in order to detect the mechanical damages. However, this technique is mostly suggested for the detection of emerging cracks [7,8]. Electromagnetic (EM) methods, such as Ground Penetrating Radars (GPR), consist of the propagation and proceeding of an electromagnetic wave into the structure to evaluate the volumetric water content through the estimation of the dielectric permittivity [9,10]. The water and chloride ion contents both contribute to the attenuation of the wave's amplitude (affected

by the dielectric permittivity of the media), but these two indicators can be difficult to distinguish separately [11] to determine the corrosion risk of rebars in early stages. Moreover, the carbonation of concrete [10] and the size of aggregates [12] can affect the value of the dielectric permittivity.

Structural Health Monitoring (SHM) techniques are another category of inspection method used in Civil Engineering that have recently attracted considerable attention [13]. In fact, the significance of SHM techniques is to provide real time and continuous monitoring, through embedded tools, in order to evaluate some specific in-situ characteristics of structures [14,15]. These technologies must remain in the structure for a long time, and such developed systems, combined with signal processing methods, allow for the detection of local or global defects [16]. Wireless sensors are just one SHM technique, and this kind of advancement of communication technologies is not representative of all types of SHM tools. Among the most reputed SHM techniques, the optic fiber Bragg-grating is used as a strain or temperature sensor that can be coupled with steel rebars to monitor geometrical changes induced by corrosion [17], or piezo-ceramics sensors are used to detect internal cracks through the measurements of a sensor's voltage [18]. However, these technologies only supply information about the initiated corrosion of steel rebars. RFID sensors know a strong interest as embedded technologies that use radio frequency (RF) or magnetic field changes for communication, and some specific and promising architectures—specifically developed for the monitoring of steel corrosion in concrete structures—exist [19,20]. However, the current drawback of these technologies is the need to provide a sufficient energy source to keep them working and to obtain the desired information. The use of RF signals in recent wireless network-based systems may be sensitive to a change of permittivity due to water ingress in concrete [20], and obviously, the interconnection between these tools requires a good network. The size of these technologies can also make their integration inside concrete structures challenging. A reliable preventive technique to evaluate the chloride ions level, which are the most important aggressive agents that would accelerate corrosion process, remains an urgent need for civil infrastructure managers.

To answer to this problem, the French ANR project LabCom OHMIGOD aims to develop a new embedded and passive SHM technology, currently under development and patentability, which is dedicated to the detection of aggressive agents diffusing in the first layer of the cover concrete area. Referred to as Functional Magnetic Material (FMM), this technology consists of the measurement of a non-destructive Magnetic Observable (MO) of a sensor embedded into the first layer of the cover concrete area of a reinforced concrete structure, and this MO varies with the presence of chlorides.

In this paper, we present the result of an experimental parametric study carried out with samples out from the concrete media to validate the concept of this technology. Indeed, the accuracy and engineering of this technology require a precise control of expected influential parameters, such as a corrosion indicator, as well as the geometry or the positioning of the FMM. This precise control is necessary to highlight a link between these parameters and the variation range of the MO that is measurable with an external interrogator. This experimental parametric study is then carried out while assuming the hypothesis: there is no influence of concrete and the environment, both identified as non-magnetic media, on the measurement of the MO. Thus, the objective of this parametric study is focusing on the physical parameters of the FMM to validate the theoretical concept of this technology and to insure that the technology can have a sufficient accuracy.

The Section 2 details the structure and the physical concept of this tool, as well as examples of applications with expected results. In Section 3, we show the results of a concise preliminary numerical study, using the Finite Element Method, carried out to show the influence of geometrical and positioning parameters on the values of the MO and to illustrate some of the physical phenomena involved in the FMM. Based on these results, Section 4 presents the objectives, the materials (prepared through controlled accelerated corrosion), the devices, and the experimental setups used for the experimental parametric study. The obtained experimental results are shown and discussed in the Section 5.

## 2. Physical Concepts of the Functional Magnetic Material Technology

The theoretical concept behind the FMM technology consists of the measurement of a non-destructive Magnetic Observable (MO), which is un-affected by the environment, quality of soil, and presence of water inside the concrete, with each of these having no magnetic characteristics. The idea is to embed a sensor, inside a concrete structure exposed to chloride, which generates an MO affected by the presence of chlorides. Figure 1 represents a picture and a scheme of an example of FMM technology that consists in two parts. The first one is a permanent part that has a cylindrical shape, with a diameter in the range between 8 to 15 mm and a thickness in the range between 2 to 5 mm, and it is generating a magnetic flux density (symbolized with $\vec{B}$) as a non-destructive MO that is measurable with a dedicated device, such as a magnetometer. The second one is the reactive part (or patch) coupled to the permanent part. The patch is a squared plate with a length at least 60% greater than the permanent part's diameter and a thickness in the range between 0.1 and 2.0 mm. It is positioned so that the upper face of the permanent part is centered on its lower face. The material of the patch is ferromagnetic, with a relative magnetic permeability estimated to 50, based on characterization results not shown in this paper, obtained from a ferromagnetic material available in our laboratory (we cannot precisely describe its nature for patentability purposes).

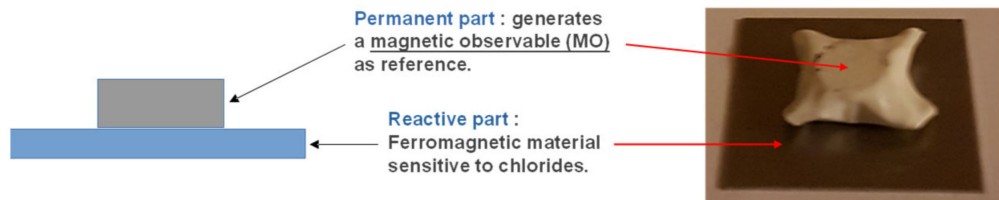

**Figure 1.** Schematic representation of a single Functional Magnetic Material (FMM).

Some research works show that ferromagnetic materials [21]—or high permeable alloys known as mu-materials [22]—can protect electronic devices from magnetic fields, using the term "magnetic shielding". The magnetic shielding consists of applying plates, made of the aforementioned materials, around zones that are sensitive to magnetic fields. Fundamentally, the ferromagnetic shields do not block magnetic fields, but they provide a path for the magnetic field lines around the shielded area [23]. In the case of mu-materials, the magnetic field is drawn into the material, which diverts it away from the shielded region [22]. The efficiency of these shields depends on their geometry but also on physical characteristics, such as magnetic permeability [21], and materials with higher magnetic permeability having better shielding properties. Thus, a patch made of a ferromagnetic material would partially shield (or filter) the MO generated by the permanent part, as the first characteristic.

The second characteristic of this patch is its sensitivity to aggressive agents, especially chloride ions, leading to its corrosion. This corrosion theoretically affects its physical characteristics (geometry and magnetic permeability), which would alter its shielding property and, therefore, the measured MO.

Figure 2 illustrates the principle of this technology through a flow chart. We assume an FMM, in its initial state, is embedded at a given depth in the cover concrete area. A magnetometer, used as an external interrogator, allows the measuring of a constant partially shielded MO. As soon as chloride ions diffuse in the porous network of the concrete, the patch will be directly in contact with them during an imbibition period and start to corrode at an initiation period. The evolution of the corrosion of the patch will lead to some morphological and physical properties changes with time, thus altering the shielding properties. It will then result in a progressive increase in the MO of the FMM with the evolution of the corrosion of the patch, which would provide an indication about the presence of aggressive agents in the area where the FMM is embedded.

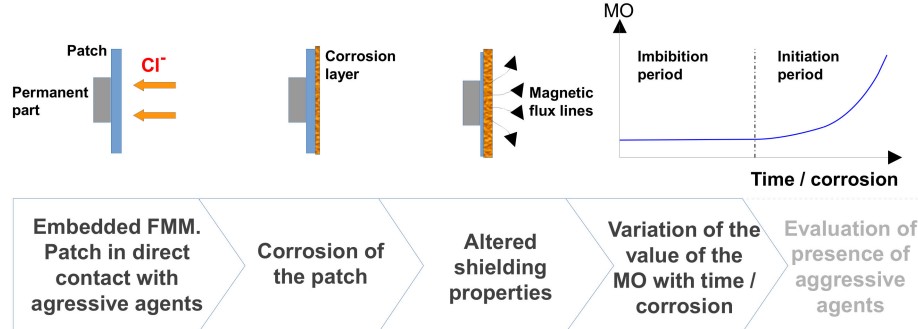

**Figure 2.** Flow chart of the principle of the FMM.

Figure 3 illustrates this concept in greater depth with an example of expected uses and signals generated by this technology. All situations in this figure are purely hypothetical and illustrate possible and expected effects. As represented in Figure 3a, FMM 1, FMM 2, and FMM 3 are embedded at different depths in the first layer of the cover concrete area of a RC structure. One can measure their respective MO with a magnetometer, displaced along the desired axis, and determine a reference value corresponding to their initial state at a time labeled as $T_0$. We assume that this structure is subjected to tides, as represented by the blue double-arrow in Figure 3a. At time, labeled $T_1$ and $T_2$, two different events occur and affect some of these FMM individually, and $T_3$ corresponds to an extended amount of time during which the corrosion, initiated by the previous events for concerned FMM, continues. The Figure 3b represents the expected changes of MO, measured with the magnetometer for each FMM, as a function of time, according to these situations.

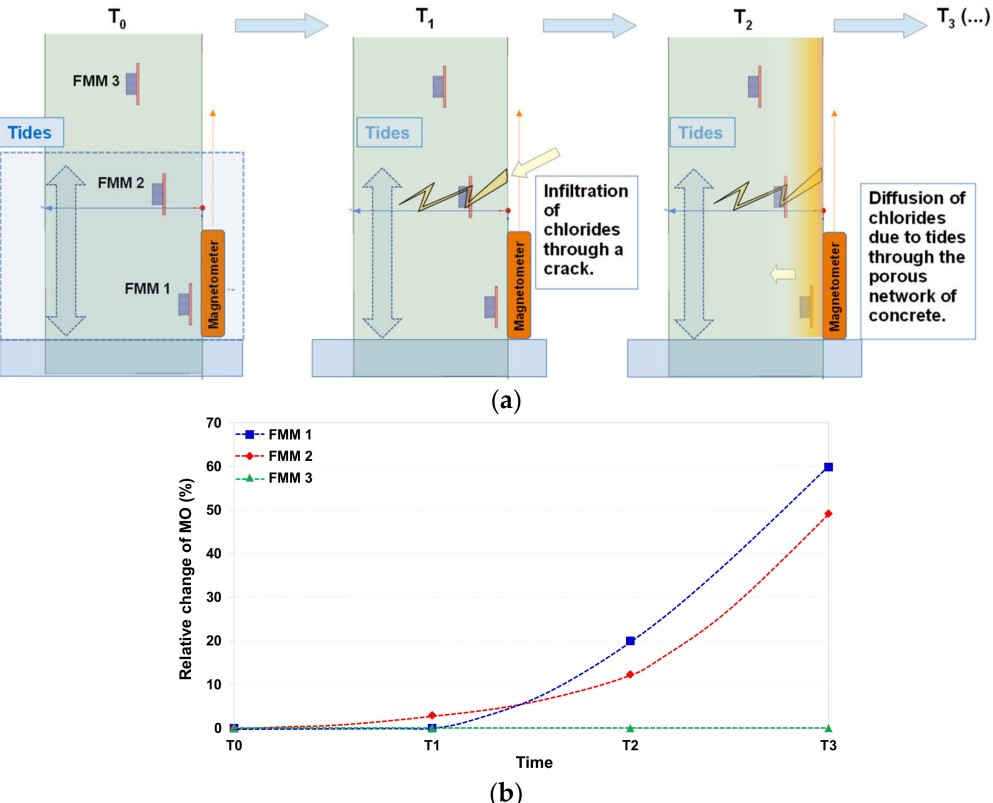

**Figure 3.** (**a**) Representation of three FMM devices—embedded at various depths in a reinforced concrete structure—subjected to tides at $T_0$, with a crack inducing corrosion of the FMM 2 at $T_1$, an additional aggressive agents' front diffusing through FMM 1 at $T_2$, and a continuation of induced corrosion phenomenon at $T_3$. (**b**) Arbitrary representation of expected changes for the MO of each FMM, as a function of time.

As shown in Figure 3a, after an arbitrary amount of time $T_1$, we assume that a crack raises and leads to the propagation of aggressive ions (due to the tides) through the FMM 2, initiating the corrosion of the patch, which starts altering its shielding properties. After a second amount of time $T_2$, we assume that some aggressive ions diffuse in the concrete and are in touch with the FMM 1, which starts to corrode, while that of the FMM 2 continues from $T_1$ to $T_2$. As represented in Figure 3b, the value of the MO, of each FMM, is relatively unchanged at $T_0$. At $T_1$, the MO of the FMM 2 increases due to the corrosion of the patch initiated by the crack and in accordance with the principle described in Figure 2. At $T_2$, the MO of the FMM 2 continues increasing due to the prolonged exposure to aggressive agents, and the MO of the FMM 1, exposed to chlorides, starts increasing. Then, as the corrosion still occurs from $T_2$ to $T_3$, the MO of these two FMM continues varying, as well. As chloride ions are never in touch with the FMM 3 in the represented situations, its MO remains unchanged. The lower value of the relative change for the FFM 2 at $T_2$ is due to its greater embedding distance compared to the FFM 1.

Hence, through a regular monitoring of the MO from each of the FMM, it could be possible to detect and follow the aggressive agents' front levels as a function of time and depth (distance between the magnetometer and the FMM). This technology appears, then, as a preventive ND method to detect the corrosion pathology in the cover concrete area before its emergence on rebars. The innovation of this technology is that it does not focus on the corrosion of rebars but, rather, of embedded materials. Furthermore, this FMM technology does not require a power source and can remain for a long time within a structure, as the permanent part generates its own signal, making it a passive technology. This MO would also, theoretically, remain un-affected by water content, granulates size, porosity, environment, temperature, or quality of the soil opposite to ND methods, based on the dielectric permittivity measurements previously mentioned. The FMM would find applications, in the domain of Civil Engineering, for the real-time and continuous monitoring of RC structures, and each FMM could provide local information about pathogen's level.

This paper focuses on a preliminary experimental validation of the concept of the FMM technology by highlighting a link between a corrosion state (or an indicator) of the patch and the MO variation range. The aim is to show, through a parametric study carried out with samples out of the concrete media (allowing a better control of influential parameters), the possibility of exploiting the variations of the shielding properties of the patch, due to its corrosion, in order to follow the corrosion fronts inside the cover concrete. Indeed, the aforementioned hypothesis about the non-magnetic characteristics of concrete and the environment enables one to clear out experimental tests in concrete mixing and to focus on a corrosion indicator on the geometry and positioning of the FMM. In Section 3, we first show the results of a concise preliminary numerical study, by using the Finite Element Method, carried out to show the influence of the patch's thickness, as well as the measurement distance, on the values of MO and to illustrate some physical phenomenon involved in the FMM.

### 3. Preliminary Numerical Results

A theoretical validation of the concept of the FMM technology was carried out through a numerical study. This section summarizes the main results, obtained and taken into consideration, to carry out the experimental study detailed in Section 4. All simulations are run through the 3D module in the Ansys software (student version), which uses the Finite Element Method (FEM) to solve the Maxwell equations. Being limited to a maximum amount of 64,000 elements for a 3D volume, we set the maximum meshing size with respect to the geometry of each elements that we simulate (4.0 mm for the permanent part and 1.5 mm for the squared-shaped reactive part). The software divides the total modeled volume into small sub-regions in order to run, both separately and for each of them, the Maxwell equations. The optimal mesh size is found, after seven iterations, in auto mode. The geometry, the material, and the relevant physical properties of each elements

were set to represent the FMM, as described in the Figure 1. For these simulations, the permanent part is set as a cylinder, with a diameter of 15 mm and a thickness of 3 mm, that generates a magnetic flux density ($\vec{B}$) and is chosen as MO. The patch is set as a squared plate, with a constant surface of 25 mm × 25 mm and a thickness set to vary between 0 and 2 mm with 0.1 mm steps. As we aim to design the FMM technology as a preventive tool, the embedding depths must be in line with the chloride concentration profiles for a better prognostic of the beginning of the initiation period (see Figure 2). Based on chloride profiles in the concrete, according to the literature [24], three distances are chosen as eligible depths at which the FMM could be embedded into the cover concrete. Thus, we perform some simulations to calculate the norm of $\vec{B}$ $\left( \|B\| = \sqrt{B_x^2 + B_y^2 + B_z^2} \right)$, chosen as MO, at points located at 1.0, 2.0, and 3.0 cm, respectively, from the center of the upper face of the permanent part along the $z$-axis of the simulated FMM, as pictured in Figure 4a. Figure 4b represents the variation of the value of the MO (norm of $\vec{B}$) as a function of the thickness of the patch for each of the three measurement points.

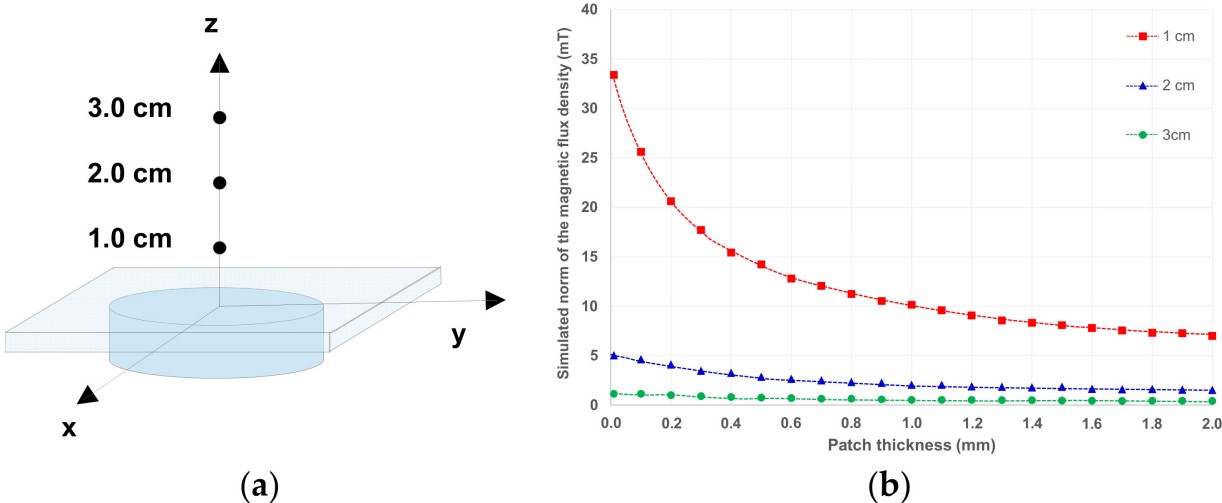

(a)  (b)

**Figure 4.** (**a**) Representation of the three measurement points located at 1.0, 2.0, and 3.0 cm from the center of the upper face of the permanent part along the $z$-axis of a simulated FMM; (**b**) values of the norm of the magnetic flux density, as a function of the patch thickness, for each of the three measurement points.

The results of this simulation show that the value of the MO tends to decrease with the increase in the patch's thickness and with the increase in measurement distance. This first result illustrates the fact that the ferromagnetic patch acts as a magnetic shield, and its efficiency increases with its thickness. For the measurement point located at 1.0 cm, the attenuation starts with a strong slope for thicknesses comprised between 0.1 and 0.5 mm, and this slope becomes nearly flat for thicknesses greater than 1.2 mm. The same tendency is observed for the other measurement points with lower values. For a measuring distance of 2 cm, the simulated values of the magnetic flux density, calculated with a 0 mm and a 2 mm-thick reactive part, are, respectively, 5.02 and 1.53 mT. These values drop down to 1.14 and 0.41 mT for a measuring distance of 3 cm. These results indicate that the technical challenges will consist of an accurate control of the distance between the FMM and the interrogator, as well as the use of sensitive and high-resolution external interrogators for the MO monitoring.

For a better understanding of these results, we choose to represent, in Figure 5, the iso-value mapping of the magnetic field obtained from three cases: (a) the permanent part alone, (b) the permanent part coupled with a 0.5 mm-thick patch, and (c) the permanent coupled with a 1.0 mm-thick patch.

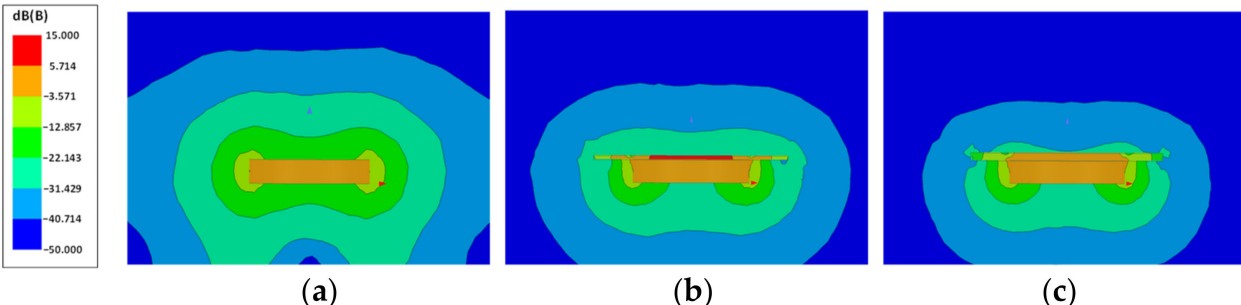

**Figure 5.** Simulated iso-value mapping of the magnetic flux density of a FMM in the case of (**a**) cylindrical permanent part alone, (**b**) permanent part coupled with a 0.5 mm-thick ferromagnetic patch, and (**c**) permanent part coupled with a 1.0 mm-thick ferromagnetic patch.

The Figure 5a shows the iso-value mapping of the magnetic flux density of a magnetic source alone. Obviously, the value of $\vec{B}$ (or the measured MO) tends to decrease as one gets further from the magnetic source along the vertical axis. The magnetic flux is greater at the edges of the permanent part. The permanent part is made of dipoles that interact and oppose each other. Thus, the field is greater near the edges because there is less magnetic dipoles that can oppose the contribution of the nearest ones.

In the case of Figure 5b, the MO measurable above the upper face of the applied 0.5 tmm-thick patch are strongly attenuated, as compared to those shown in Figure 5a. This attenuation is slightly greater with the increase in the patch's thickness, up to 1.0 mm (Figure 5c), which confirms the tendencies observed in Figure 4b. Greater magnetic flux densities exist inside the patch and, especially, in the regions close to the center and the edges of the permanent part (red, orange, and yellow zones in Figure 5b,c). A reasonable interpretation of this result is that the coupling between the ferromagnetic patch (with a high magnetic permeability) and the high magnetic flux densities at the center and the edges of the permanent part generates local zones with high magnetization. These zones deviate the magnetic flux lines of the permanent part, leading to a decrease in the MO measured above the patch. The greater the thickness for the patch, the stronger the contribution of these high magnetization zones, which attenuate the MO along the vertical-axis.

As the patch would corrode in the presence of chloride ions, this would alter this part of the FMM in term of geometry and probably physical properties. Based on results shown in Figures 4 and 5, one can ask if the geometrical variation of the patch (or the alteration of its physical properties) could lead to an increase in the MO measured from the FMM. This is the objective of the experimental study detailed in this paper. In the next section, the experimental setups and protocols, as well as the materials and devices used to carry out the experimental study, are described.

## 4. Devices, Experimental Setups and Materials

The aim of this paper is to show, experimentally, that the corrosion of the patch affects the MO of an FMM. To demonstrate this claim, we carry out an experimental parametric study to measure the value of the MO of FMM samples, made with a reference permanent part, coupled with patches that have controlled corrosion indicators based on their Relative Mass Loss (RML) [25]. As shown by the numerical study, the thickness of the patches affects the value of the measured MO. Therefore, in a first step, we prepared some patches with two different thicknesses and imposed some specific RML to some of them through an accelerated corrosion process by electrolysis. In the second step, each of the prepared patches are separately coupled with a reference permanent part, and the value of the MO for each such-obtained FMM samples is measured with a magnetometer. As the FMM are designed to be embedded at different depths into the cover concrete, this parameter is set by stacking, between the samples and the magnetometer, some PVC plates (concrete and PVC have both negligible magnetic susceptibilities [26,27]) in order to measure the value of

the MO as if the samples were embedded at different depths into concrete. As shown in the numerical study, the distance between the FMM and a magnetometer strongly affects the value of the MO. We chose to use two magnetometers with different detection ranges and resolutions. The first magnetometer is a gaussmeter with a hall probe that has a high detection range and a relatively low resolution that is suitable for measurement at close distances. The second magnetometer is a 3D-magnetometer with a lower detection range but a better resolution, which is dedicated for measurement at higher distances. For each of these magnetometers, a specific experimental setup is used. The preparation of the patches and the description of the two magnetometers with their associated experimental setups are, respectively, detailed in the following sub-sections.

### 4.1. Samples Preparation

Based on results shown in Section 3, the width of patches are 60% greater than the diameter of the permanent part. As all magnetic sources available in our laboratory have an 8 mm diameter, the patches have a fixed surface (squared shape 15 mm × 15 mm) from ferromagnetic material plates with 0.5 and 1.0 mm-thicknesses.

As shown in the literature [25], the corrosion rate of materials can be evaluated in terms of mass loss per unit of time. Therefore, we chose to set a Relative Mass Loss (RML) as the corrosion indicator of our patches. These RML are set, through an accelerated corrosion method, by electrolysis in a prepared 30 g·L$^{-1}$ NaCl solution (equivalent to the chlorides concentration of seawater). The patches are separately marked, weighed, and then immersed in the NaCl solution contained in a beaker. The prepared patches are linked to the positive terminal of an electrical generator and considered as anodes, while a titanium electrode coated with platinum, linked to the negative terminal, is used as a cathode. The polarity of these electrodes remains unchanged during this whole process. The Figure 6 show a scheme and a picture of this setup.

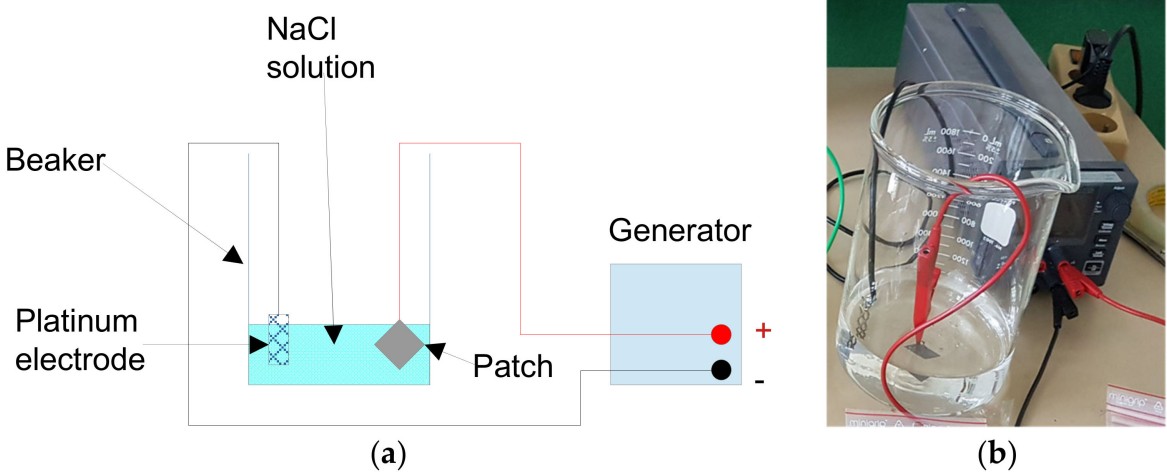

**Figure 6.** (**a**) Scheme and (**b**) picture of the accelerated corrosion setup.

With the use of the first electrochemical law of Faraday (given by $M = I. z. t$), and with the application of a constant electrical intensity of 10 mA with a generator, it was possible to evaluate the duration of corrosion in order to set a specific RML for each patch. $M$ is the mass loss (grams), $I$ is the electric intensity (amperes), $t$ is the corrosion duration (seconds), and z is the electrochemical constant depending of the material's molar weight (gram·mol$^{-1}$), its electronic valence, and the Faraday's constant (96,485 coulomb·mol$^{-1}$).

Figure 7a illustrates the series of the 15 mm × 15 mm and 1.0 mm-thick patches before and after corrosion. Similarly, the Figure 7b displays the 0.5 mm-thick patches.

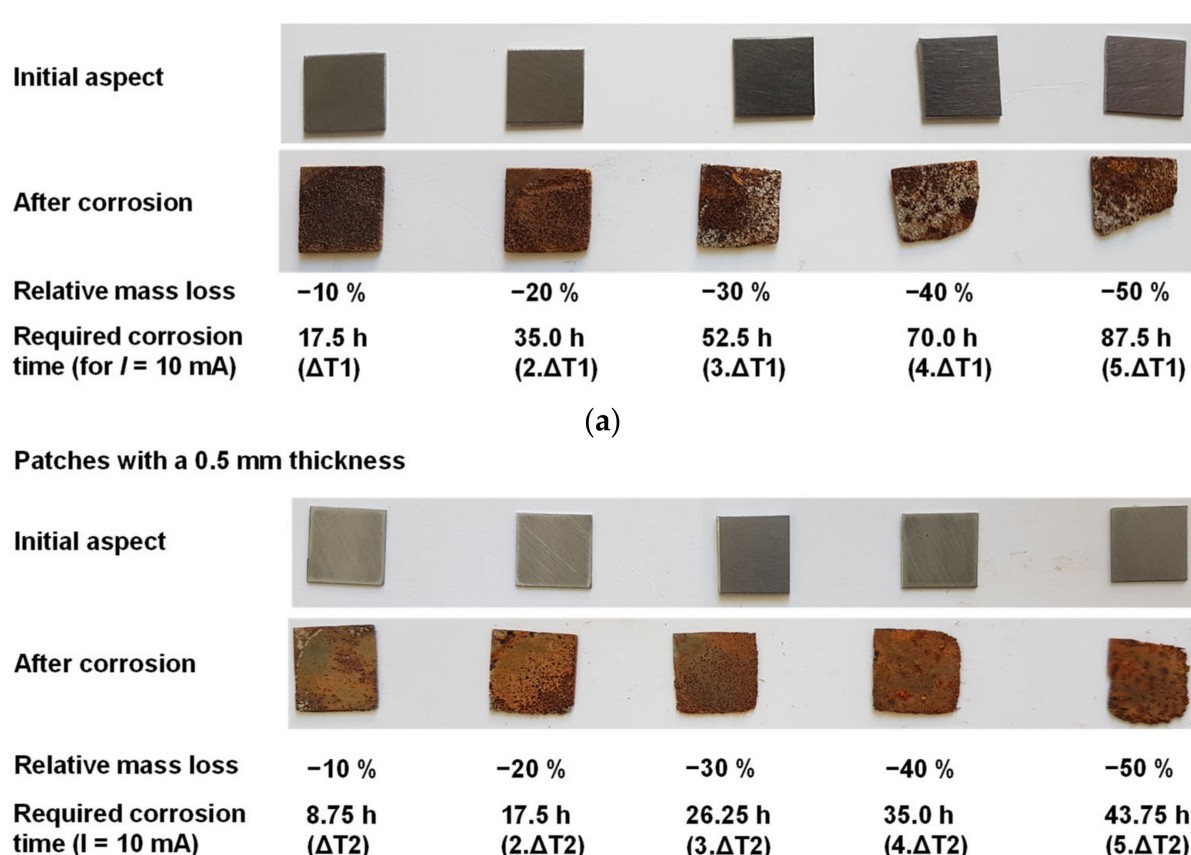

**Figure 7.** Aspect before and after the accelerated corrosion of five patches with (**a**) 1.0 mm thickness and (**b**) 0.5 mm thickness. The expected Relative Mass Loss (corrosion indicator) and the required time of accelerated corrosion to reach it are indicated below each sample.

For these two batches, five different RML are separately set between 10% and 50%, with 10% steps for each patch. These percentages are arbitrarily chosen to represent an indicator of corrosion for the patch, as a function of time, after a sufficient exposure to chloride ions and to test the accuracy of the FMM technology. For the 1.0 mm-thick patches, each 10%-RML step requires a duration $\Delta T1 = 17.5$ h of accelerated corrosion; this amount of time is reduced by a half for the 0.5 mm-thick patches ($\Delta T2 = 8.75$ h). The expected RML are reached with errors in the range of $\pm 0.5$%-RML. For the two available thicknesses, a patch is set aside and kept in its non-corroded (or initial) state.

The pictures show that the corrosion starts from the corners of patches, and the material losses become more significant with the increase in the RML. For the 1.0 mm-thick patches, an alteration of the shape occurs, starting from 30% of RML, and leads to significant material losses with higher RML. Obvious material losses are present for the 0.5 mm-thick patches after 50% of RML. Using a digital caliper (precision of $\pm 0.01$ mm), the thickness of the patches linearly lowers in slight proportion with the increase in RML: a loss of thickness in the range of 0.08 mm was measured for the 1.0 mm-thick patch with 50%-RML. We have chosen to neglect these variations.

All prepared patches are separately coupled to a single reference permanent part (assigned for all measurements of the experimental study), so the center of their respective upper and lower faces were aligned along the *z*-axis (parallel along the patches' thickness). We design such assembled materials as FMM samples. By convention, we use the term "patch corroded at XX%" to mention a patch that has a RML of XX% (XX = 10 to 50%). The

term "non-corroded patch" mentions a patch that has not been corroded (remains in its initial state). Subsequently, we separately measure the generated MO from each sample with two magnetometers by taking into consideration the influence of the thickness of the patch, the RML of the patch, and the measurement distance (between the sample and the magnetometer). All measurements were taken in a room far from any external magnetic source (no magnetically shielded chambers were available in our lab). Sections 4.2 and 4.3, respectively, describe the characteristics and experimental setups associated with these two magnetometers.

### 4.2. Hall Gaussmeter

A Hall gaussmeter GMH104 (CAYLAR S.A.), pictured in Figure 8a, is primarily used. This device is equipped with a Hall probe that allows one to measure the magnetic component values, i.e., the magnetic flux density in the direction of $\hat{x}$, $\hat{y}$, and $\hat{z}$ (respectively, $B_x$, $B_y$, and $B_z$) of a magnetic source. The maximum detection range of this magnetometer is ±20,000 gauss (G) with a resolution of 0.1 G. The values of $B_x$, $B_y$, and $B_z$ are displayed, in real time, on the digital screen of the gaussmeter and recorded in a computer with a dedicated acquisition software.

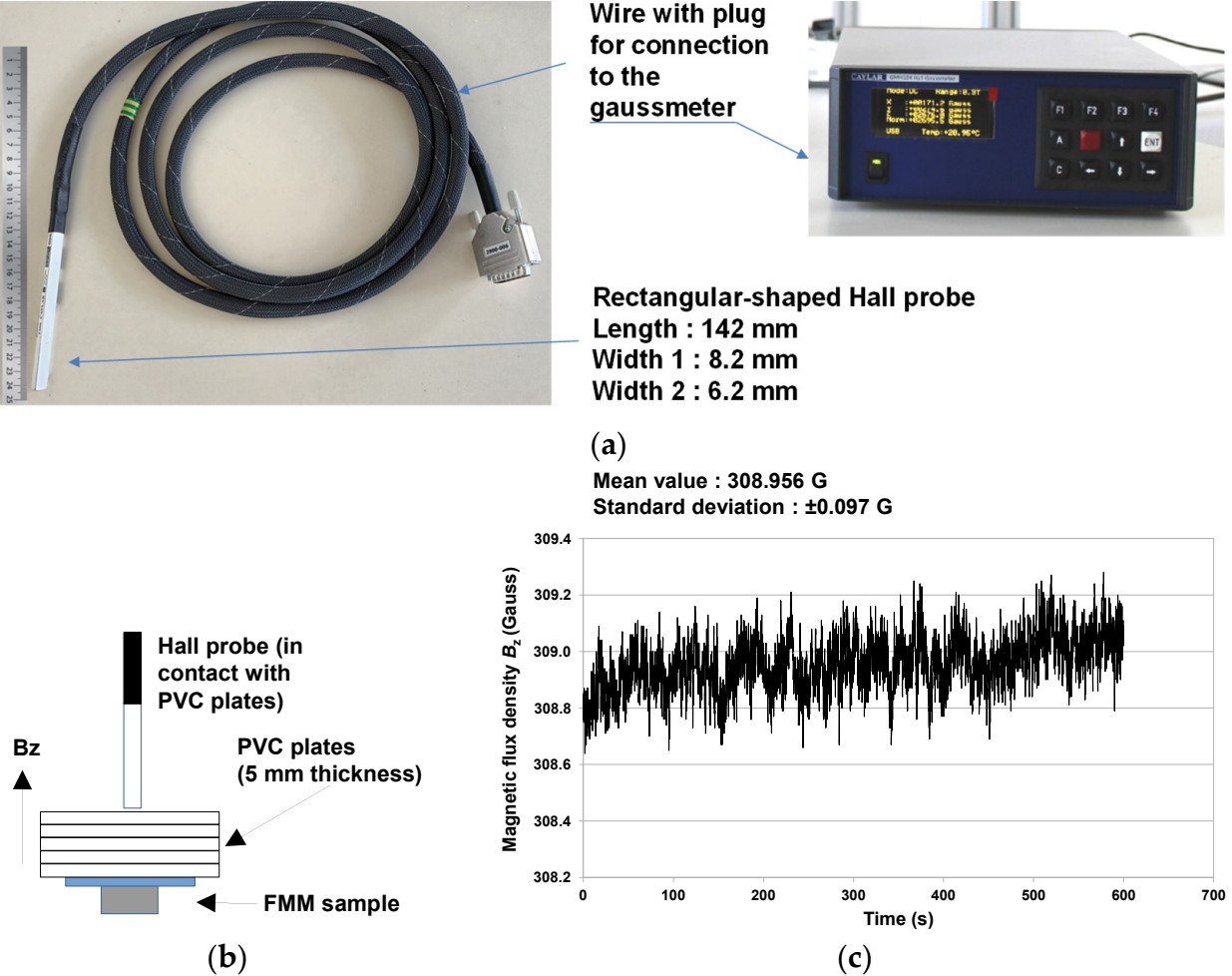

**Figure 8.** (**a**) Pictures of the Hall probe and gaussmeter with the sizes of the probe, (**b**) schematic representation of the experimental setup used for the measurement of $B_z$ with the Hall probe, and (**c**) example of values of $B_z$, recorded during 10 min with the Hall gaussmeter, from the reference magnetic source.

Figure 8b illustrates the experimental setup dedicated to the use of the gaussmeter. The FMM samples (made of the reference magnetic source coupled with a patch) are

placed under stacked 5 mm-thick PVC plates, and consequently, the Hall probe is placed in contact with the top plate. The aim of this experimental setup is to measure the MO of a sample, as a function of a theoretical embedding depth, set by PVC plates. The PVC is a diamagnetic material as its magnetic susceptibility is negative, in the range of $-1.1 \times 10^{-5}$ (SI) [26], while concrete materials are identified as paramagnetic with a positive magnetic susceptibility in the range of $1.3 \times 10^{-3}$ SI [27]. Considering the range of values for the magnetic flux densities measured, these susceptibilities can be neglected. The distance range between the Hall probe and the sample is arbitrary set from 10 mm to 35 mm (with 5 mm steps considering PVC plates' thickness), which represents eligible embedding depths for the FMM, as mentioned in Section 3. The direction of measurement is in $z$-axis direction, thus, the component of $B_z$ is the most desired MO, in this case. In this setup, the amount of the other components, $B_x$ and $B_y$, were close to zero. Therefore, each measurement is associated with the position of the probe through the vertical direction of the $z$-axis.

We have performed a test with the reference permanent part to check the stability and the representativeness of the measurements. The same operator realized all measurements. As we carried out all measurements in a room that was not magnetically insulated, we first reset the bias values of the Gaussmeter far from any magnetic source and metallic materials. It allowed us to eliminate any deviations possibly caused by the environment (magnetic field of Earth, temperature ... ). Then, we place the probe above our reference permanent part and have recorded the values of $B_z$ during 10 min (at a frequency of 10 Hz) with the acquisition software. The hall probe is manually maintained above the reference magnetic source, the recorded signal is plotted in Figure 8c. We have extracted a mean value of 308.9 G, with a standard deviation ($\sigma$) close to 0.1 G from these data. The increasing tendency observed may be associated to a slight displacement of the probe during the manual measurement. As this stability test is considered as satisfying, the value of $B_z$ of each sample could be recorded during only 1 min, with an uncertainty lower than 0.1 G, and in order to extract a representative mean value of $B_z$ and $\sigma$. The value of 0.1 G, deduced from the 10 min-measure, is considered as representative of the uncertainties of repeatability.

The experimental procedure with the gaussmeter consists of measuring the values of $B_z$ of FMM samples for each distance set, from 10 mm to 35 mm, with 5 mm steps. We realize the first measurements with the permanent part alone. These values of $B_z$ are labeled as reference values. Next, the permanent part is coupled with the non-corroded patches (one of each available thickness). These values of $B_z$ are mentioned as initial values of $B_z$ and will correspond to the MO measured from a FMM sample in its non-corroded (or initial) state. Finally, the permanent part is coupled with each of the 10 corroded patches, separately (Figure 7a,b), to assemble corroded FMM samples. Then, we compare these values to the reference values and the initial values of $B_z$. Some complementary measurements are realized with a 3D-magnetometer, which has characteristics and an associated experimental setup that are detailed in the next paragraph.

*4.3. 3D Magnetometer*

The second device used for the experimental study is an MTi-630 3D-magnetometer from X-sense [28], as pictured in Figure 9a. This device uses an Anisotropic Magneto Resistance (AMR) technology that allows for measuring the values of $B_x$, $B_y$, and $B_z$ of a magnetic source up to 8 Gauss in absolute value. According to the constructor's data-sheet, this 3D-magnetometer has a resolution of 0.25 mG, a total RMS noise of 1 mG, and a linearity error of 0.2%.

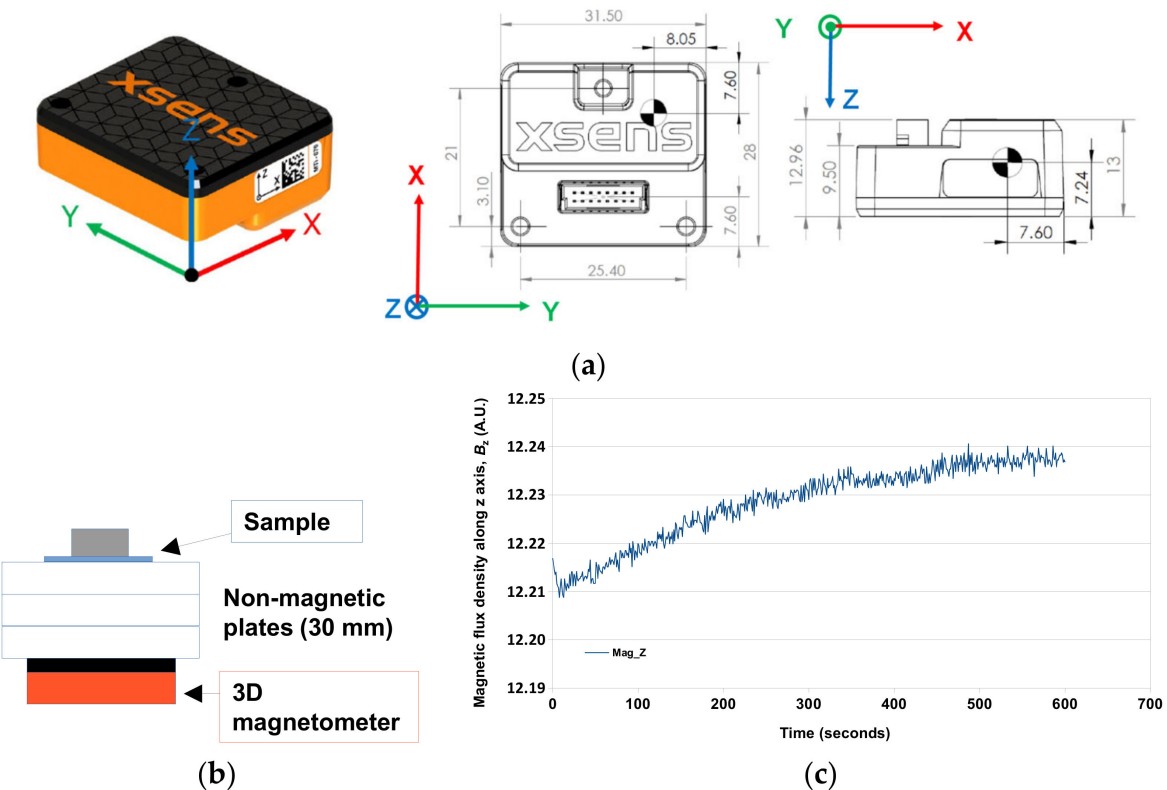

**Figure 9.** (**a**) Picture and details about the 3D magnetometer (reprinted from reference [28]), (**b**) schematic representation of the experimental setup used for the measurement of $B_z$ with the 3D-magnetometer, and (**c**) example of values of $B_z$ recorded during 10 min with the 3D-magnetometer from a reference magnetic source.

The Figure 9b is a scheme of the experimental setup used specifically with the 3D-magnetometer. Some non-magnetic plates are stacked above the 3D-magnetometer in order to set a distance of 30 mm between the top surface of the 3D-magnetometer and a FMM sample (positioned above the plates) with the patch oriented downward. With the use of an acquisition software displaying, in real-time, the values of $B_x$, $B_y$, and $B_z$, the samples are positioned so that $B_z$ is maximized, while $B_x$ and $B_y$ are close to zero. The samples and the 3D-magnetometer are not touched during the recording. The values are displayed in Arbitrary Units (A.U.), with 1 A.U. corresponding, approximately, to 0.5 G, according to the constructor's data-sheet.

Similar to the previous paragraph, we have also realized a stability test, during the 10 min with the reference magnetic source, to check the representativeness of recorded data. The Figure 9c shows the obtained curve representing the value of $B_z$ (chosen as MO for this experiment) as a function of time (recording frequency: 10 Hz). A linear deviation occurs during the five minutes of recording, and then, $B_z$ starts to stabilize. We attribute this deviation to a hot processing of the 3D-magnetometer that occurs during each acquisition. With this device, the acquisition duration for each sample is 10 min, and then, the mean values and standard deviations of $B_z$ are extracted from the last minute of acquisition.

The experimental procedure with the 3D-magnetometer consists of, first, recording the value of $B_z$ for the permanent part, alone, to obtain the reference value for a measurement distance set to 30 mm. Next, the permanent part is coupled with each of the 10 patches shown in Figure 7 in their non-corroded state. The aim is to measure the initial value of $B_z$ for each of the patches before their corrosion to check the sensitivity of the 3D-magnetometer. Each of the patches are marked and corroded, according to the process described in Section 4.1. Then, the permanent part is coupled with each of the corroded patches in order to highlight the influence of the RML of a patch on $B_z$, compared to the

reference value and the initial value of $B_z$. We show and discuss these results in Section 5.2, in line with the results obtained from the Hall gaussmeter detailed in Section 5.1.

## 5. Results and Discussion

### 5.1. Measurements with Hall Gaussmeter

Figure 10 represents the mean values of $B_z$ measured with the Hall gaussmeter as a function of distance (comprised between 10 to 35 mm) for the permanent part, alone, and coupled with the 1.0 mm-thick patches (corroded and non-corroded). The standard deviation, σ, for each measurement done with the Hall gaussmeter, is 0.1 G, and the error bars are not represented for readability purposes. The distance ranges are arbitrarily chosen as eligible embedding depths for the FMM based on chlorides concentration profiles shown in the literature [24].

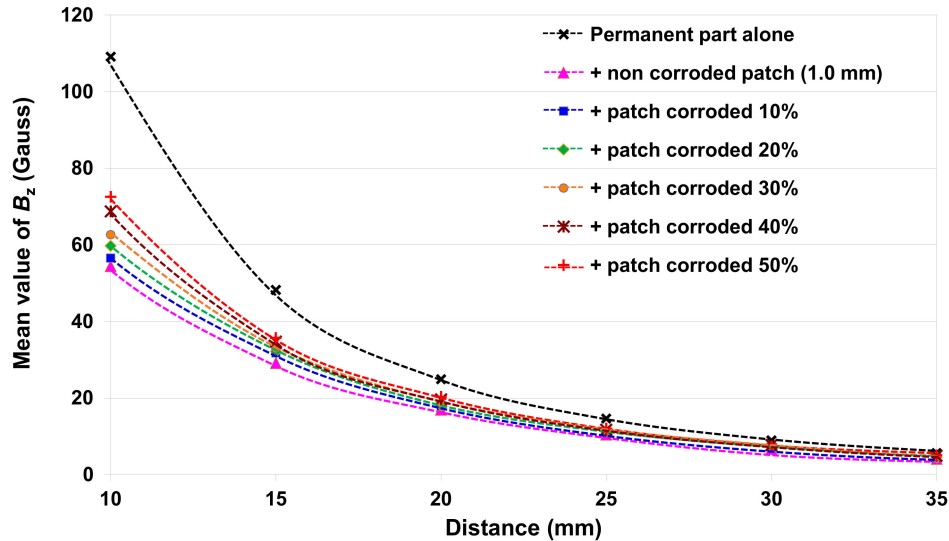

**Figure 10.** Mean value of $B_z$ measured with the Hall gaussmeter as a function of measurement distance for the permanent part, alone and coupled with 1.0 mm-thick patches with different RML, from the 10 mm to 35 mm range.

The value of $B_z$ obviously decreases with the increase in the measurement distance. For each distance, the reference value ($B_z$ measured from the permanent part alone) is greater than the initial value of $B_z$ (permanent part coupled with a non-corroded patch), which shows that the coupled reactive parts attenuate the MO generated by the permanent part. Compared to the initial value, we observe that, with the increase in the RML, $B_z$ rises as well. These results illustrate a lowering of the shielding properties of the patch with corrosion, which validates, experimentally, the concept of the FMM technology.

All else being equal, higher mean values of $B_z$ are measured from samples coupled with 0.5 mm patches (not shown in this paper) compared to those coupled with 1.0 mm-thick patches. These observations are in line with the results shown in Figures 4 and 5 and discussed in Section 3. The higher thickness of the patch enhances the contribution of high magnetization regions responsible of the deviation of the MO, which lowers the measured MO. Moreover, as shown in Figure 7, some losses of materials have occurred, in great proportion, for the 1.0 mm-thick patch corroded at 50%. One of our current assumptions is that the geometry of the patch is a parameter to consider for the magnetic deviation phenomenon involved in the FMM. Some additional specific characterizations could be carried out in order to check if the corrosion process alters the magnetic physical properties of the patch (for example, the magnetic permeability/susceptibility) and quantify their influence on $B_z$. The homogeneity of the corrosion may have to be considered, as well.

For a better analysis about the effect of the RML and distance on the variation of the FMM's MO, we calculate the relative change of $B_z$ ($\Delta B_z$ in %), considering the value from the

permanent part, alone, as the reference. For a given distance, $\Delta B_z$ is calculated, according to Equation (1), from the reference value for the permanent part alone ($B_{z\ \text{Permanent part}}$) and the value of $B_z$ measured for the permanent part coupled with a patch with a given RML ($B_{z\ \text{Permanent part+patch}}$):

$$\Delta B_z = \frac{B_{z\ \text{Permanent part}} - B_{z\ \text{Permanent part+patch}}}{B_{z\ \text{Permanent part}}}, \tag{1}$$

We also chose to calculate the uncertainty for $\Delta B_z$ ($\sigma_{\Delta B_z}$), according to Equation (2). We calculate a new quantity, based on Equation (1), by adding the positive value of the standard deviation of the Hall gaussmeter ($\sigma$) to $B_{z\ \text{Permanent part}}$ and by subtracting $\sigma$ to $B_{z\ \text{Permanent part+patch}}$. The difference between this quantity and $\Delta B_z$ corresponds to the calculated $\sigma_{\Delta B_z}$.

$$\sigma_{\Delta B_z} = \frac{\left(B_{z\ \text{Permanent part}} + \sigma\right) - \left(B_{z\ \text{Permanent part+patch}} - \sigma\right)}{\left(B_{z\ \text{Permanent part}} + \sigma\right)} - \Delta B_z, \tag{2}$$

Figures 11 and 12 represent the relative attenuation of $B_z$, as a function of the distance for each sample, respectively constituted with the 1.0 mm-thick patches and the 0.5 mm-thick patches in their non-corroded and corroded states. The labels indicate the values of $\Delta B_z$, measured at 10 mm and 30 mm, from the sample coupled with the non-corroded patch (upper label in pink) and the patch corroded at 50% (lower label in red), with their respective calculated uncertainty.

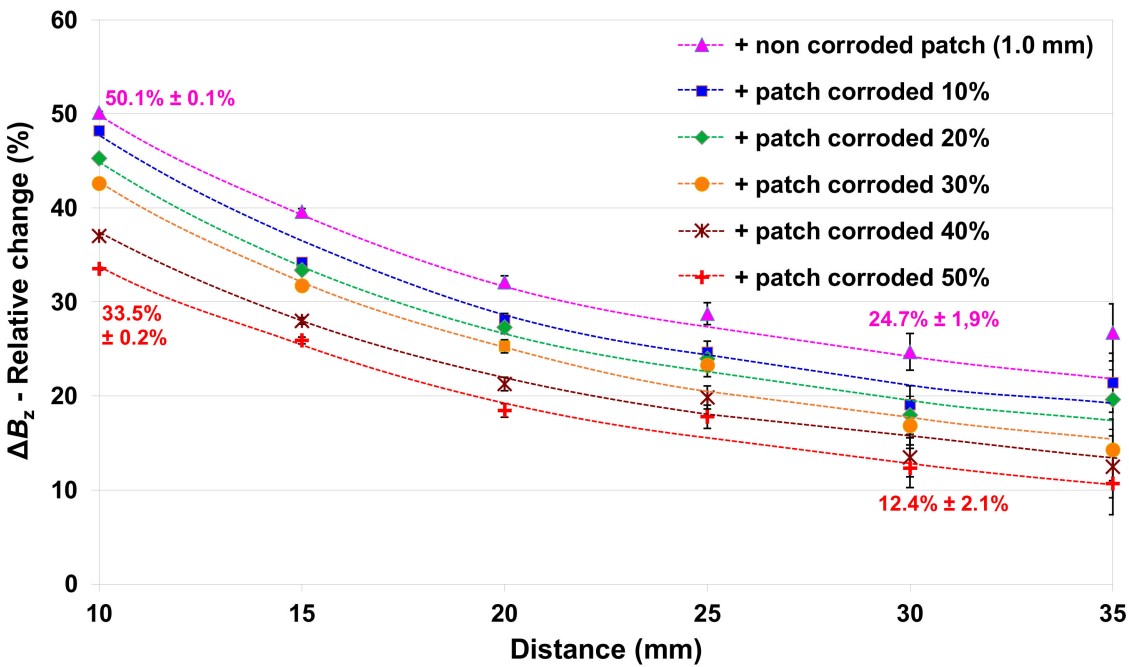

**Figure 11.** Relative change of $B_z$ as a function of distance for samples constituted with the 1.0 mm-thick patches.

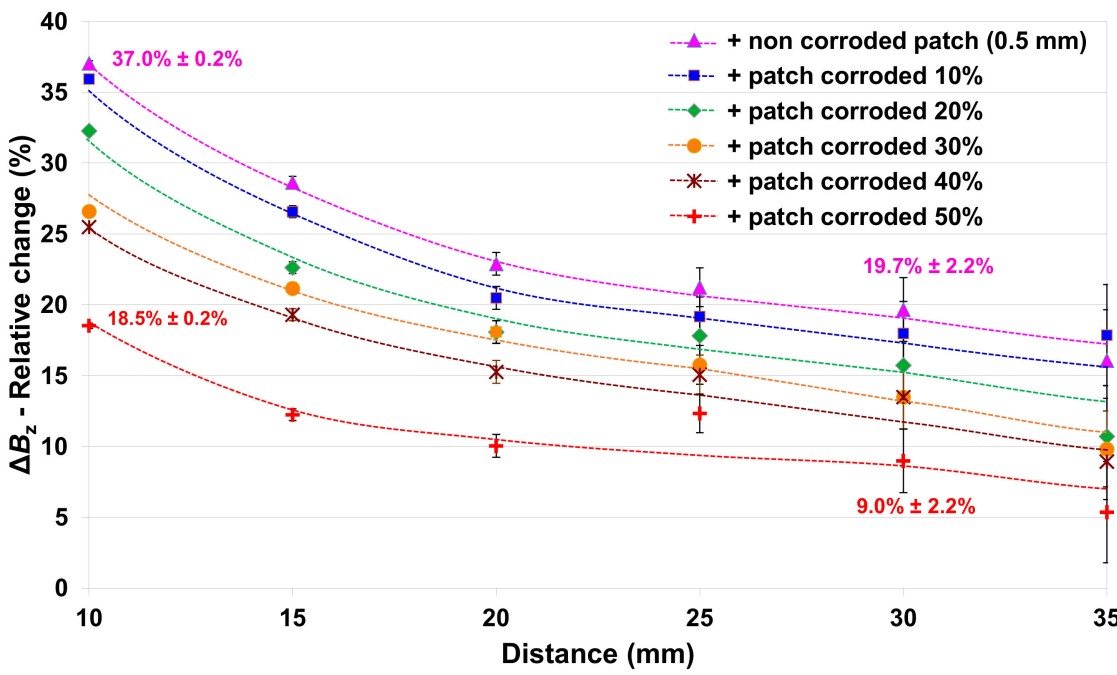

**Figure 12.** Relative change of $B_z$, as a function of distance, for samples constituted with the 0.5 mm-thick patches.

For all cases, we observe a decrease in the relative change with the increase in distance. The slope of all curves tends to be smoother with the increase in distance. These observations indicate that the shielding effect of the patch is more efficient at a close distance. For a fixed distance, $\Delta B_z$ tends to decrease with the increase in the RML of the patches. We interpret this as a loss of efficiency of the shielding property of the patches, which is consistent with the results shown in Figure 10 and validates the principle of the FMM technology. These results also indicate that the MO of the FMM technology will start to vary when the RML of the patch is close to 10%, and this change would be measurable at different depths in the cover concrete region.

For a distance of 10 mm, considering the 1.0 mm-thick patches (Figure 11), the $\Delta B_z$ for the samples, coupled with the non-corroded patch and the patch corroded at 50%, are, respectively, 50.1% and 33.5%, with respective uncertainties of 0.1% and 0.2%. In the case of the 0.5 mm-thick patches (Figure 12), these values are, respectively, 37.0% and 18.5% (with uncertainties of 0.2%). This difference between these two extreme values could represent, quantitatively, the accuracy of the FMM that seems greater at a short distance.

For a distance of 30 mm, lower values of $\Delta B_z$ are obtained. In the case of the non-corroded and the 50%-corroded 1.0 mm-thick patches, their respective $\Delta B_z$ drop down to 24.7% and 12.4% (Figure 11), while the values measured from the 0.5 mm-thick patches are, respectively, 19.7% and 9.0% (Figure 12) but with uncertainties in the range ±2.0%. Indeed, due to the resolution of the Hall gaussmeter and the decrease in the mean values of $B_z$ with distance, the uncertainty rises up as well. This indicates a loss of accuracy with the increase in embedding depths for the monitoring of $B_z$. We cannot discuss the results obtained from the measurements realized at 35 mm due to an even higher uncertainty increasing up to ±3.6%. These results also show that the thickness of the patch influences the FMM accuracy; this point is more deeply discussed in the Section 5.2.

Thus, through a regular monitoring of the FMM sensors embedded at relatively short distances (lower than 30 mm in our case), it would be possible to evaluate the RML of the reactive part as a function of $\Delta B_z$. As the embedded reactive part would corrode in the presence of chlorides, the FMM technology could be used as a nondestructive monitoring tool for the detection of chloride ingress front. Theoretically, the FMM technology would open opportunities to follow the progression of a chloride front at different depths in the

cover concrete. Due to the decrease in $\Delta B_z$ at higher distances (greater than 25 mm), the use of magnetometers with better resolutions may be required, as shown by the results obtained with the 3D-magnetometer in the Section 5.2.

### 5.2. Measurements with the 3D-Magnetometer

In complement with the results obtained from the Hall gaussmeter, and due to high uncertainties for distance greater than 25 mm, some additional measurements are proceeded with the 3D-magnetometer (described in the Section 4.3), with a lower detection range but a better resolution. Following the experimental procedure previously described, the obtained results are represented in Figure 13.

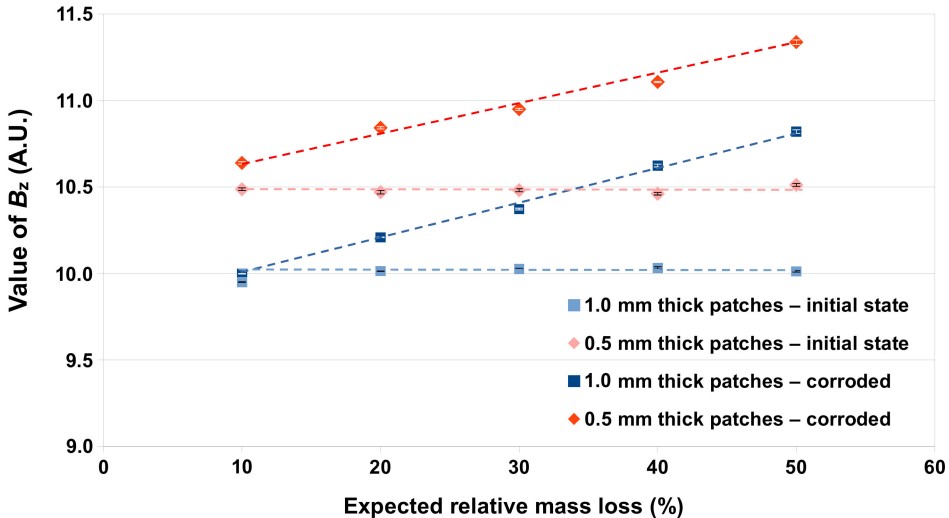

**Figure 13.** Mean value of $B_z$ measured with the 3D-magnetometer of each sample before and after corrosion and at a distance of 30 mm; clear marks that correspond to measurements of non-corroded samples are placed on the expected relative mass loss axis so that their value of $B_z$ before and after corrosion could be compared.

We remind that, for this experimental setup, we have fixed the measurement distance from the top surface of the 3D-magnetometer and the characterized sample to 30 mm. The value of $B_z$ measured from the permanent part, alone, was 12.243 ± 0.003 A.U., and it is considered as the reference value for the calculation of $\Delta B_z$. The clear marks represent the mean value of $B_z$ measured from samples coupled with each of the non-corroded 0.5 and 1.0 mm-thick patches. The standard deviations for each of these measurements are comprised between 0.001 and 0.014 A.U. in absolute value. The slight differences observed between the values of $B_z$ may be attributed to some manual positioning errors of the samples or to some inaccuracies of patches' dimensions due to the mechanical cuttings. As the values of $B_z$ are relatively close for each of the five patches with the same thickness, the data obtained with the 3D-magnetometer can be considered as representative. The dark marks correspond to the values of $B_z$ measured from samples coupled with the corroded patches. The value of $B_z$ tends to increase linearly with the RML of patches, and a change of $B_z$ can be detected starting from 10%-RML. The values measured from the samples, coupled with the 0.5 mm-thick patches, are greater than those with the 1.0 mm-thick patches. These results show that the monitoring of FMM sensors embedded at 30 mm may still be possible with a magnetometer that have an adapted resolution and detection range.

Using the Equations (1) and (2), as well as the standard deviations extracted from the measurements, $\Delta B_z$ and the uncertainty are calculated and plotted as a function of the expected RML in Figure 14.

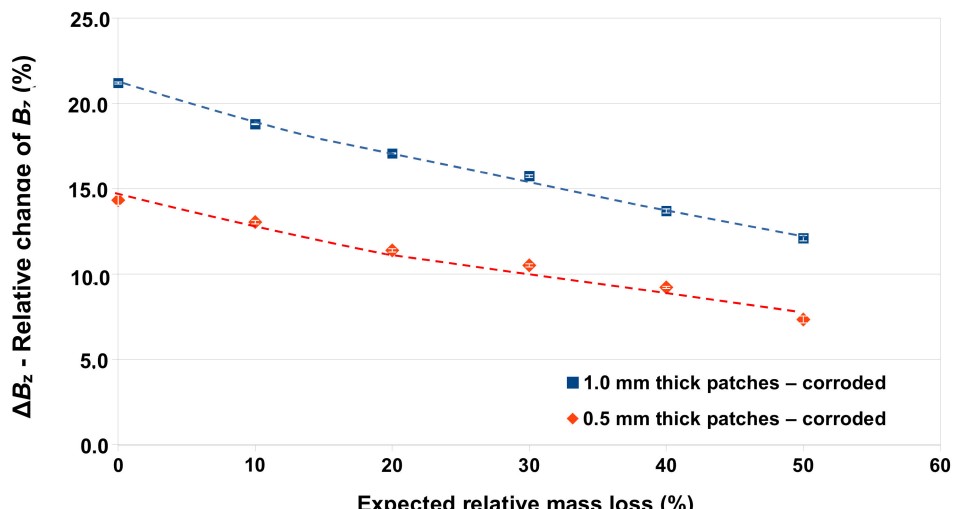

**Figure 14.** Relative change of $B_z$, measured from the 3D-magnetometer, as a function of the relative mass loss of patches.

In line with previous results, $\Delta B_z$ decreases with the increase in the RML of the patches. In the case of samples coupled with 1.0 mm-thick patches, from 0 to 50%-RML, respectively, $\Delta B_z$ drops down from 21.2% to 12.1%, which represents a global difference of 9.1%. These changes are in equivalent ranges as those plotted in Figure 11, for measurement distances comprised between 30 to 35 mm and for the same considered RML range.

The value of $\Delta B_z$ is lower with samples coupled with 0.5 mm-thick patches and the obtained values being 14.3% and 7.3% for respective 0 and 50%-RML; this represents a global difference of 7.0%, which is lower than the one obtained from 1.0 mm-thick patches. To discuss this point, addressed in the Section 5.1, we chose to plot, in Figure 15, the value of $\Delta B_z$—of each sample—as a function of their respective applied accelerated corrosion durations (see Figure 7).

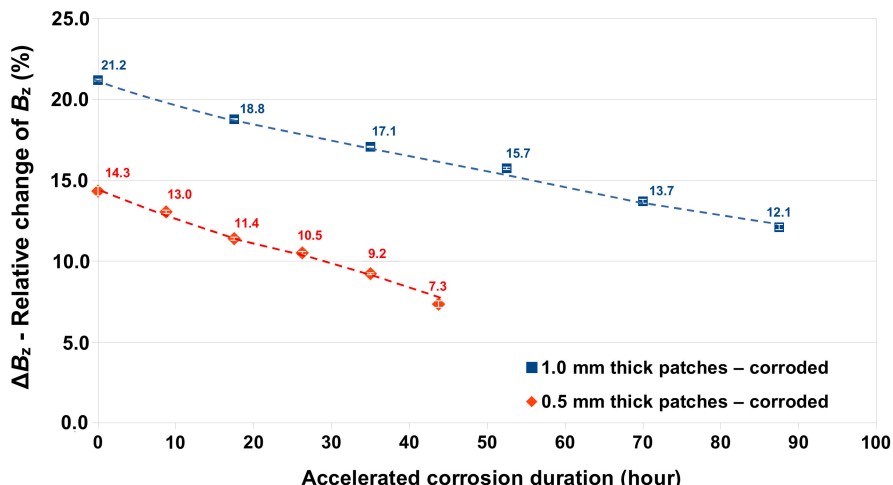

**Figure 15.** Relative change of $B_z$, measured from the 3D-magnetometer, as a function of the accelerated corrosion duration applied on 0.5 and 1.0 mm-thick patches.

Assuming a constant corrosion rate (mass loss per unit of time) for patches inside concrete, the RML of 1.0 mm-thick patches will be half those of a 0.5 mm-thick patch (see the accelerated corrosion duration labeled in Figure 7). Thus, for an amount of time, making a 0.5 mm-thick patch corrode at 40%, the RML of a 1.0 mm-thick patch will be 20% for the same time exposure. The global difference from 0 to 20%-RML is 4.1% for the 1.0 mm-thick patch and 5.1% from 0 to 40%-RML for the 0.5 mm-thick patch, meaning that a FMM made

with a thin patch will be slightly more sensitive. The use of a thin patch for the FMM technology will then, theoretically, allow an earlier detection of chloride ingress but with a lower associated $\Delta B_z$ compared to those measured from 1.0 mm-thick patches. This means that a compromise has to be found between the geometry of the FMM and the embedding depths, as well as the specificity of the magnetometer, to follow the diffusion of a chloride front in RC structures.

### 5.3. Discussion and Prospects

As claimed in the introduction, the study of the influential parameters of this FMM technology requires processing of measurements with samples out from concrete media. We also assumed that this media has no influence on the MO. The final validation of this technology will, indeed, require a further experimental campaign, with samples embedded into concrete media. This further campaign would have two independent objectives: firstly, checking that the reactive part can corrode inside concrete and that the change of MO is measurable with a dedicated interrogator (our current study nearly validate this point); the second point would be to study the link existing between the reactive part corrosion and the chloride ingress. Such relations would be of major interest for infrastructure managers, but they would be the subject of another dedicated study. However, as it stands, this technology is still under development, and some additional numerical and experimental studies are required to optimize its accuracy. The influence of other parameters for the patch, such as smaller or greater surfaces and thicknesses, the geometrical shape and some physical properties, such as the magnetic permeability, should be considered to contribute to a better understanding of the magnetic flux lines deviation phenomenon occurring in the patch and how they influence the measurement of $B_z$. The use of a permanent part generating a greater/lower MO may also affect the relative attenuation, as well as the accuracy, of the FMM. Moreover, a study of the kinetics and mechanisms of a natural chloride-induced corrosion of the patches, as a function of the aforementioned parameters, would provide information to proceed to accurate simulations and experiments to confirm the obtained results.

We can also check the influence of other parameters, such as the chloride concentration inside some concrete slabs positioned between the sensors and the magnetometer or the magnetic properties of corrosion products. The temperature and the pH inside concrete can also be considered. One can intuitively assume that these parameters would have a negligible effect on the MO measurements, but a dedicated study will clear any doubts.

As different operators can proceed to the measurement of the MO of embedded FMM, this could bring uncertainties due to a different positioning of the external interrogator along the *x*- or *y*-axis. Based on our experiments, these uncertainties are greater at short distances. Thus, for a regular and precise monitoring of the MO, a way to accurately position the interrogator, with respect to a relative position of an embedded FMM, is required.

The main aim behind these studies is to set an analytical model that allows estimating, separately, for various FMM embedded at different depths in a RC structure, a specific non-destructive magnetic characteristic affected by chlorides concentration. The Figure 14 could be considered as an example of calibration curves allowing evaluating the RML of the patch from $\Delta B_z$. The same curves could be extracted from the results shown in Figures 11 and 12. If we can establish a relation between the MO and an estimated concentration of chlorides in a further work, a probability of contamination level by aggressive agents, in the areas where the FMM are embedded, can be introduced as a new preventive durability indicator.

### 6. Conclusions

Chloride ions are the most important aggressive agents diffusing in the porous network of a reinforced concrete (RC) structure and that accelerate the corrosion process of steel rebars. To answer the need in Civil Engineering of a reliable preventive technique to detect the chlorides before they reach the steel reinforcement, we present in, this paper, a parametric study to optimize and demonstrate the concept of a new passive SHM

technology, mentioned as Functional-Magnetic Material (FMM). It aims to evaluate the chloride ions level in the first layer of the cover concrete area of RC structures and to detect a front of aggressive agents in order to prevent the corrosion of steel reinforcements. The concept of the FMM is based on the measurement, with a magnetometer, of a Magnetic Observable (MO), generated by the permanent part and partially shielded by the reactive part (or patch), which has characteristics (geometry, physical properties) that are altered by a chloride-induced corrosion.

This concept is preliminarily validated through a numerical study, where the results show that a ferromagnetic material, used as a patch, can partially shield the MO due to magnetic coupling phenomena, which deviate the magnetic flux lines generated by the permanent part. Hence, the geometry of the patch (that should have a surface at least 60% greater than the permanent part one's) and the distance along the $z$-axis are identified as influential parameters leading to a decrease in the magnetic flux density along the $z$-axis ($B_z$), chosen as MO.

We have carried out an experimental study with samples out of concrete media to validate these observations and the concept of the FMM technology. In this study, we have, indeed, assumed the hypothesis that the concrete media has no influence on the measurement of the MO, allowing us to focus on the identified influential parameters. FMM samples, made of a reference permanent part coupled with patches that have controlled Relative Mass Loss (RML, chosen as the corrosion indicator obtained through chloride-induced electrolysis) are prepared and their MO measured with two different magnetometers. The influence of the patches' thickness and RML, as well as the distance between the FMM and the magnetometer (corresponding to eligible embedding depths at with the FMM could be embedded in concrete), are considered.

At a fixed distance between the FMM and a magnetometer, the experimental results show that the relative change of the MO (based on the reference value measured from the permanent part alone) tends to decrease with the increase in the RML of the patch. These results illustrate the possibility to monitor the MO of a FMM embedded at a given depth, which validates, experimentally, the concept of this technology. The experimental results also confirm that the MO of the FMM decreases with the distance, and the relative changes induced by the RML of the reactive part decrease as well. Whatever the distance, some significant relative variations of MO can be measured when the patch loses at least 10% of its initial mass. Hence, an early detection of chlorides ingress would be possible. The use of a thin patch increases the accuracy of the FMM, but the relative change of the MO based on the reference value is lower, which would require sensitive magnetometers.

This experimental parametric study is a first necessary step for the preparation of a further experimental campaign involving samples embedded into concrete. Our results show that the control of the influential parameters, validated in this study, is necessary, but an optimization is required to accurately follow the corrosion fronts inside the cover concrete. Our current results suggests that, through a regular monitoring of the MO (and more precisely the relative change), it would be possible to evaluate and to follow, as a function of time and depth, the RML of the reactive part, linked somehow to the presence of aggressive agents. For FMMs located at distances greater than 30 mm (in the studied experimental conditions), the use of high-resolution magnetometers may be required for a precise monitoring of the MO. The FMM technology appears as a preventive nondestructive monitoring tool with a strong potential for a preventive detection of chlorides ingress in the first layer of the cover concrete area of a RC structure.

**Author Contributions:** Conceptualization, A.I., D.G. and X.D.; methodology, D.S. and S.K.; validation, D.S., A.I., X.D., D.G. and S.K.; formal analysis, D.S.; investigation, D.S. and S.K.; data curation, D.S.; writing—original draft preparation, D.S.; writing—review and editing, A.I., X.D. and S.K.; visualization, D.S., A.I., X.D., D.G. and S.K.; supervision, A.I. and X.D.; project administration, A.I.; funding acquisition, A.I. All authors have read and agreed to the published version of the manuscript.

**Funding:** This research was funded by the French "Agence Nationale de la Recherche" (ANR) through the project 18-LCV2-0002 (https://anr.fr/Projet-ANR-18-LCV2-0002 accessed on 21 July 2022).

**Data Availability Statement:** Not applicable.

**Conflicts of Interest:** The authors declare no conflict of interest. The funders had no role in the design of the study; in the collection, analyses, or interpretation of data; in the writing of the manuscript, or in the decision to publish the results.

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
