# Peer review of "Experimental Parametric Study of a Functional-Magnetic Material Designed for the Monitoring of Corrosion in Reinforced Concrete Structures"

_remotesensing, doi:10.3390/rs14153623_

Round 1
Reviewer 1 Report
The article discusses current issues in the field of electromagnetic non-destructive testing of materials. the topic of the article is relevant, the results are presented in an understandable way. It is necessary to make some corrections and changes in the article - please see my recommendations and comments.
Fig. 4 and further: function values ​​should not be connected by a line unless we clearly know the course of the function between adjacent points. use approximation, interpolation, extrapolation, splines... What is the value of B on the vertical axis? Is it a module or an absolute value of one of the components?
paragraph 4: As the FMM are designed to be embedded at different depths into the cover concrete, this parameter is set by stacking between the samples and the magnetometer some PVC plates (concrete and PVC are non-magnetic materials) in order to measure the value of the MO as if the sample was embedded at different depths into concrete. Q: what does the statement that it is a non-magnetic material mean? Is it a diamagnetic or paramagnetic material?
part 4.1: you state that you immersed the samples in saline solution. But this does not correspond to the used salt concentration of 30g/l. It is necessary to correct this data. Or it is necessary to correct the information whether it was a solution with parameters of sea water or a physiological solution. Furthermore, it is necessary to add information whether the polarity of the electrodes was changed during the electrolysis process.
Based on Fig.7, the following time intervals are sufficient for the purposes of the experiment: 17.5hrs, 35hrs, 70hrs and possibly 140hrs. so the coefficient between the intervals is equal to 2.
part 4.2: How was the repeatability of the magnetic field measurements ensured? The question is what external interfering fields could have been present during the Hall gaussmeter measurement and whether they could have been negligibly small. Were the measurements taken in a magnetically shielded chamber?
Fig. 8a: instead of the image on which nothing is clearly visible, I recommend inserting a new image - a detail of the Hall sensor, with a scale /for example, a ruler/.
The same applies to Fig. 9: please provide the detail of the AMR sensor. Operating electronics is not the subject of the article.
In Figures 10 and 11, the authors show waveforms whose shape can be easily predicted based on theoretical knowledge. it is an exponential decrease in the intensity of magnetic fields with increasing distance from the sample. this is well known. instead of Fig. 10a and Fig. 10b, I recommend showing such images: relative intensity changes - related to one reference sample. It is clear from the above images that the strength of the magnetic field is decreasing and the differences between the individual curves are getting smaller - and this can be assumed.
Throughout the article, it is necessary to unify the style of labeling quantities: for example, electric current is labeled using the NORMAL font and should be italic. The magnitudes of the magnetic field are correctly given in italics, but the subscripts should be NORMAL. This needs to be fixed in the graphs as well.
It is necessary to unify the type of decimal separator throughout the article.
Fig. 15 and further: the name relative variation is not correct. Variations are a specific term used e.g. in music theory. In the graph it is a change, I recommend using the name of the axis: relative change....
Reviewer 2 Report
The subject of the article is very interesting and deserves to be valorized. Here are some remarks to further increase the value of the document.
Lines 104-105
“The Figure 1 represents a picture and a scheme of this FMM technology that consists in two parts.”
Provide here (not in Section 3) indication on the dimensions of the two parts and indicate the corresponding quotas in Figure 1.
Figure 3
Is there any theoretical reason why you estimated that the effect of chloride diffusing along a crack is less than the effect of chloride diffusing by porosity, or this is just a schematic representation of the possible effects? Please provide qualitative or quantitative information on the expected variations in Figure 3(b) (in the main text).
Furthermore, as far as the effect of the crack is concerned, it is reasonable that gravity plays a role in the diffusion of the chloride, along with the surface tension. Can you provide some comments on this aspect?
Line 206
“By applying a proper meshing size with respect to the geometry of each elements”
Provide information on the mesh size considered appropriate for the given elements.
How was the “proper evaluation” performed by the software: in closed form or iteratively? In the latter case, what optimum condition did the software use and how many iterations did it take to get the optimal size?
Figure 5
To allow direct comparison, use the same scaling for the three images: enlarge Fig. 5(a) to the same image scaling as Fig. 5(b) and Fig. 5(c).
Round 2
Reviewer 1 Report
The authors have included all comments and suggestions in the article. After the revision, the article is homogeneous, with a clear informational value. it will certainly be an interesting professional enrichment for the interested reader.
Reviewer 2 Report
The authors adequately addressed all issues raised by the reviewer.
This manuscript is a resubmission of an earlier submission. The following is a list of the peer review reports and author responses from that submission.
Round 1
Reviewer 1 Report
Remarks and Questions
- Fig. 1 is not clear.
- Is the material of the patch representative of the corrosion process ?
- What are the magnetic properties of the corrosion products ?
- The reading distance is a key parameter for the application. The distance of 3 cm is sufficient ?
- It seems difficult to validate a sensor for monitoring of corrosion in reinforced concrete structures with no result in concrete.
Synthesis
This paper reports the development of a passive sensor for corrosion detection in concrete. The topic is of first interest. However, since a direct application in concrete is not proved in the manuscript, a journal dedicated to sensors could be more suitable. In the introduction, the authors propose to develop an embedded sensor in the concrete cover. Similar strategies were recently proposed by different groups and should be more detailed. In particular, the application of the RFID technology to this problem should be developed. The results are very promising. However, some additional information should be given for the reader. In particular, the physical basis responsible of the shielding is not detailed. The influence of magnetic permeability and electrical conductivity on the shielding should be discussed more thoroughly with additional simulations.
Author Response
Greetings,
We, the authors, thank you for the reviewing of our paper and have well received your remarks and suggestions.
Please find attached with this message a Word document entitled "Cover letter with answers to reviewers". This file is divided in three sections, the two first corresponding respectively to the answers we brought to your remarks and questions, as well as these of the second reviewer, and a third section listing all major changes proceeded.
We remain available to supply any information you may consider required/necessary.
Sincerly yours.

Reviewer 2 Report
The submitted paper provides the work on experimental validation of a Functional-Magnetic Material designed for the monitoring of corrosion in reinforced concrete structures. This type of work is interesting when the preventative method of chloride ion detection in the cover concrete needs to be designed. The manuscript is comprehensive and completed with some necessary discussions. However, there are a few suggestions as follow for its usability if these issues can be satisfactorily clarified.
- The authors should carefully conduct a grammar check because some unnecessary mistakes have occurred.
- In introduction, the summary of the research status is insufficient, and the necessity of this work is not prominent enough. The specific test method design should be introduced in detail in a separate chapter, not in the introduction.
- FMM technology is mainly used to detect the chloride ion concentration of cover concrete. However, all experiments in this paper do not apply FMM technology to concrete. Therefore, this greatly reduces the reliability of the research results in this paper. It is hoped to supplement the experimental results on the application of FMM technology in concrete.
Author Response

(The authors gave the same response as above.)
